# Essential Oils and Their Natural Active Compounds Presenting Antifungal Properties

**DOI:** 10.3390/molecules24203713

**Published:** 2019-10-15

**Authors:** Maurine D’agostino, Nicolas Tesse, Jean Pol Frippiat, Marie Machouart, Anne Debourgogne

**Affiliations:** 1Laboratoire Stress Immunité Pathogènes, EA7300, Faculté de Médecine, 9 avenue de la Forêt de Haye, 54505 Vandoeuvre-les-Nancy, France; jean-pol.frippiat@univ-lorraine.fr (J.P.F.); m.machouart@chru-nancy.fr (M.M.); a.debourgogne@chru-nancy.fr (A.D.); 2Société Septeos, 12 avenue de la grande armée, 75017 Paris, France; nicolas.tesse@neteos-groupe.com; 3Service de Parasitologie-Mycologie, CHRU de Nancy, Hôpitaux de Brabois, 11 allée du Morvan, 54511 Vandœuvre-les-Nancy, France

**Keywords:** essential oil, antifungal, invasive fungal infection

## Abstract

The current rise in invasive fungal infections due to the increase in immunosuppressive therapies is a real concern. Moreover, the emergence of resistant strains induces therapeutic failures. In light of these issues, new classes of antifungals are anticipated. Therefore, the plant kingdom represents an immense potential of natural resources to exploit for these purposes. The aim of this review is to provide information about the antifungal effect of some important essential oils, and to describe the advances made in determining the mechanism of action more precisely. Finally, the issues of toxicity and resistance of fungi to essential oils will be discussed.

## 1. Introduction

### 1.1. Public Health Problem, Invasive Fungal Infections (IFIs)

Fungal infections are mainly recognized by their superficial lesions, which are often benign but sometimes difficult to treat, for example, an onychomycosis with tablet involvement. However, fungal infections can also be systemic in severe immunosuppression and in many cases, the prognosis is poor. Despite the constant progress in medical practices, invasive fungal infections (IFI) remain a major problem in France and worldwide and the incidence continues to rise (the incidence of IFI rose by 1.5 per year between 2001 and 2010) [1]. 

Between 2001 and 2010, more than 35,000 cases of invasive fungal infections were recorded in France, most of nosocomial origin, causing nearly 10,000 deaths [1].

In 2010, invasive candidiasis, which is the most common fungal infection in France (43%), was the deadliest (40.6%), and was also the fourth leading cause of nocosomial infections in the United States [2].

It affects nearly 3.6 per 100,000 persons per year and nearly 400,000 worldwide [3]. The infection starts with candidemia and may then spread to other organs [1].

Aspergillosis, the second most common fungal infection in France, affects more than 4 persons per 100,000 with a mortality rate of more than 28% [4] and it is estimated that there are almost 300,000 cases per year worldwide [5]. This high mortality rate can be explained by delays in diagnosis or management. The speed of care is therefore a crucial element for the vital prognosis (Table 1).

The incidence of other rare infections, such as mold infections (*Scedosporium* spp, *Fusarium* spp, *mucorales*) has increased over the past 20 years. These infections are often fatal in immunocompromised patients, with a mortality rate of over 60%, because of resistance to most of the drugs used in clinical practice [4,6]

Despite the technological improvements in the detection of these molds, a quick and accurate diagnosis can be difficult and the number of patients at risk continues to grow. In fact, the increase in immunosuppressive treatments (chemotherapy, transplantation, etc.) leading to severe and prolonged neutropenia is a major risk factor of developing such mycoses [7].

In addition, although the underlying diseases differ according to the species involved, the immunosuppression of patients and the use of prophylactic and empirical treatments remain the main cause of emergence of these infections. 

### 1.2. Crop Disease and Fungal Infections 

Fungal infections are not just a human health problem. They also affect the field of agriculture. In fact, it is estimated that 20% and 40% of the total agricultural productivity loss is caused by animals, weeds and pathogens. These losses have implications in human health, environment, and economy [8]. 

In the 21st century, it is estimated that the cross-loss is due to 18% animal pest and 16% microbial disease (70%–80% due to fungi), for an average loss of 68% of the crop production tonnage [9]. Among these plant pathogens we find, for example, *Botrytis cinerea*, which is responsible for the rot gray mold and can infect more than 200 species of plants (fruits, vegetables, ornamental flowers) [10]. Another example is *Fusarium graminearum* implicated in *Fusarium* head blight in wheat and with significant economic consequences, for example between 1998 and 2000, it caused an estimated loss of 2.7 billion dollars in the United States [11].

### 1.3. Current Antifungal Treatments and Therapeutic Failures

Since the discovery of amphotericin B in 1955 [12], only a decade of drugs is available in the current antifungal arsenal for the treatment of invasive fungal infections. At present, there are five classes, each with a different mode of action [13]:Polyenes, such as amphotericin B, target the plasma membrane, and in particular, ergosterol, a major component of the fungal membrane that increases membrane fluidity and causes cell death.Echinocandins, such as caspofungin, target β (1–3) glucan in the fungal cell wall.Azole derivatives, such as fluconazole or voriconazole, act on the biosynthetic pathway of ergosterol, increasing membrane fluidity and accumulation of a toxic sterol.Allylamines, such as terbinafine, also target the plasma membrane, and in particular, the first steps of ergosterol synthesis.Pyrimidines, such as 5-fluorocytosine, act on the nucleus, and in particular, on DNA synthesis.

Despite the existence of these different antifungals, hospitals in France and around the world are facing therapeutic failures in the treatment of invasive fungal infections. These failures can be explained by limiting drug interactions in patients treated for different pathologies, the severity of the infection, an antifungal concentration that is too low, or the immune status of the patient. However, cases of natural or acquired pathogen resistance are more and more frequent as the fungal mechanisms of escape from therapy involve biofilms [14].

Currently, only a few new drugs are being developed. However, the high cost and toxicity of the available drugs as well as the emergence of fungal resistance, justify the search for new classes of antifungals or innovative therapeutic strategies [15]. Resources of the plant world in terms of active substances used for therapeutic purposes are very vast and as yet, are underestimated. Some of the pharmacological properties of essential oils extracted from plants (leaves, flowers, roots or bark) are anti-infectious and antioxidant. Many classes of molecules contained in essential oils like terpenoid and phenylpropanoid give them these therapeutic activities [16,17]. 

Due to this increase in incidence, the high mortality related to invasive fungal infections in humans, and the limitations of current therapeutic strategies, new classes of antifungals are required. In this field, the plant kingdom has an immense potential for the exploitation of natural resources.

## 2. The Main Essential Oils with Antifungal Activity

The Agence National de la Sécurité du Médicament (ANSM) defines essential oils as “odorous products, generally of complex composition, obtained from a botanically defined plant raw material, either by entrainment by steam, or by dry distillation, or by a suitable mechanical process without heating”. Approximately 30,000 plants can be used to produce essential oils and more than 150 different types of oils are on the market. Known for hundreds of years, they are used in many fields, such as the food industry, perfumery, cleaning products, traditional medicine and aromatherapy [18].

For many years, numerous scientific research in human pathology has focused on essential oils in relation to their antimicrobial, anti-inflammatory and anxiolytic effects, for example.

Currently, the main essential oils studied for their antifungal activity are thyme oil, rich in thymol and carvacrol, tea tree oil rich in terpenes, and peppermint or clove oil [19], although many others have also been shown to be effective against fungi. The essential oils presented in this review were chosen for their proven antifungal effect and their frequent use. Many others exist elsewhere, but we chose to study a limited number so as not to overload this presentation.

### 2.1. Lamacieae Family 

Thyme essential oil (*Thymus vulgaris*) is already known to be effective against fungi infecting humans. Its antifungal activity is due to its high concentration of thymol and carvacrol [20].

Ahmad et al. demonstrated an inhibition of *Candida albicans* and *Candida tropicalis* with *Thymus vulgaris* essential oil and these major constituents at 62 µg/mL. Daferera et al. showed an activity on *Fusarium* spp with an ED50 (dose of essential oil that inhibits 50% of mycelium) at 71 µg/mL [21]. Finally, a study by Klarić et al. showed that molds such as *Aspergillus* spp, *Penicillium* spp and *Cladosporium* spp could be completely inhibited with a thyme oil concentration of 9.85, 19.17 and 15.20 µg/mL, respectively [22].

This oil could also play a role in agriculture. In fact, Banani showed an effect of thyme essential oil against *Botrytis cinerea*, responsible for gray mold disease, which affects most crops and causes major losses all over the world. In this study, this essential oil causes a decrease in the diameter of the wounds caused by this fungus and the overexpression of the defense genes [23]. It is also effective against *Alternaria brassicae*, capable of infecting broccoli, cabbage or rapeseed in particular, with an ED50 at 677 µL/mL, and against *Fusarium oxysporum*, a pathogenic fungus that attacks more than a hundred species of plants, with an ED50 at 363 µL/mL [24].

Finally, it also acts on *Fusarium graminearum*, responsible for fusariosis of wheat and corn, with a 100% MFC (minimal fungicidal concentration) for a concentration of 115 to 108 µg/mL [25].

Moreover, in one of these studies Pinto et al. successfully demonstrated the effectiveness of *Thymus pulegioides* essential oil against various filamentous fungi (*Aspergillus* spp and Dermatophytes) and yeasts (*Candida* spp) with MICs (the minimal concentration that inhibits 100% growth of the fungi) between 0.16 and 0.64 µg/mL [20].

Oregano essential oil (*Origanum vulgare*) has been studied for its effectiveness against bacteria responsible for food spoilage, as well as for its potential role as an antioxidant, anti-inflammatory, anti-diabetes and cancer suppressing agent. Its antifungal effect has also been recognized [26].

In addition, Daferera et al. demonstrated an efficacy against *Botrytis cinerea* and *Fusarium* sp., with an ED50 of 50 µg/mL against these two fungi [21].

A study by Santoyo et al. also found a MIC of 1.48 to 1.75 mg/mL against *Candida albicans* and 2.75 to 2.85 mg/mL against *Aspergillus niger* [27]. Finally, another study by Khosravi et al. showed a MIC between 0.5 and 1,100 µg/mL against *Candida glabrata* [28]. Although the difference between MICs for the same species is important, it is not surprising. Indeed, each strain may respond differently according to their own sensitivity.

Pennyroyal oil *(Mentha pulegium*) is studied for its antiseptic properties in the treatment of various pathologies such as colds, sinusitis or more serious diseases such as cholera, as well as for its antioxidant and antimicrobial activity. In his study, Mahboudi et al. found MICs of 1 μL/mL against *Candida albicans* and of 0.25 μL/mL for *Aspergillus niger* [29]. MICs determined between 800 and 1000 µg/mL against *Candida* spp and 222 µg/mL for *Aspergillus* spp are highlighted in the review by Stringaro et al. [30].

Peppermint essential oil (*Mentha piperita*) is already well known for its medicinal applications in the treatment of symptoms of the gastrointestinal tract such as nausea, vomiting or indigestion for example, but also for its pharmaceutical applications as a vasoconstrictor or for its wide use in the food industry [31].

Its antifungal activity has often been proven, for example, a MIC of 500 ppm completely inhibits the growth of *Candida albicans* [32], and a concentration of this essential oil between 40 and 7000 μg/mL, 800 and 3500 μg/mL and 400 and 3,500 μg/mL, completely inhibits the growth of *Candida* spp. dermatophytes and *Aspergillus* spp, respectively [30].

First known in aromatherapy for its relaxing and sedative virtues, lavender essential oil (*Lavandula angustifolia*) is now studied for its effectiveness against microorganisms, including fungi. For this oil, Daferera et al. also determined an ED50 against *Botrytis cinerea* and *Fusarium* spp of 223 μg/mL and 520 μg/mL respectively [21], and Dianez et al. showed an ED50 of 372 μL/mL against *Fusarium oxysporum* [24]. In addition, studies against *Candida albicans* have been conducted. For example, a study by Behmanesh et al. shows a significant antifungal effect of lavender essential oil after diluting the pure solution 40 times. [33] Finally, a concentration of between 0.125% and 2% *v*/*v* completely inhibits its growth [34].

Fresh or dried in dishes, rosemary (*Rosmarinus officinalis*) is also used as an essential oil for its benefits, which include antioxidant, anti-inflammatory, anti-diabetic and anti-cancer therapeutic properties [35]. This essential oil also shows antimicrobial activity against various microorganisms, including pathogenic fungi. According to Daferera et al., the ED50 determined for this essential oil is approximately 600 μg/mL against *Botrytis cinerea* and 660 μg/mL against *Fusarium* sp [21]. In addition, 24 to 31 μg/mL would be sufficient to inhibit 80% of the growth of *Candida albicans* [36]. Finally, a study by De Oliveira et al. also demonstrated a MIC of 0.78 mg/mL against *Candida albicans* [37].

### 2.2. Myrtaceae Family 

Already known for its immunomodulatory effect as well as its anti-inflammatory and antibacterial action [38], tea tree essential oil (*Melaleuca alternifolia*) also has an antifungal effect that was successfully evaluated in dermatomycological infections [39].

In fact, Carson et al. showed an effect against yeasts, dermatophytes and various filamentous fungi with species-dependent variations. For example, according to the species studied, the MIC of *Aspergillus* spp varies between 0.016%–0.12% *v*/*v* and that of *Candida* spp between 0.03%–8% *v*/*v* [39].

In addition, a study conducted on immunocompromised mice showed an effectiveness of this essential oil at 1.95 mg/mL for *C. albicans* [40].

Eucalyptus contains more than 800 species and is one of the most used plants in the world. Eucalyptus essential oil, which is mainly composed of 1.8 cineole (or eucalyptol), is studied for its antimicrobial, antiparasitic, antidiabetic, antihistaminic, anti-inflammatory, and antioxidant activity [41]. The *Eucalyptus citriodora* essential oil is the most marketed in the world, and its efficacy has been demonstrated against *Candida albicans* with a MIC of 318 µg/mL [42].

Moreover, in one of his studies, Elansany et al. demonstrated the effectiveness of *Eucalyptus camaldulensis* oil against various pathogenic fungi including *Penicillium funiculosum*, *Aspergillus niger* and *Aspergillus flavus*, and showed a MIC of 0.15 mg/mL, 0.47 mg/mL and 0.43 mg/mL, respectively [43].

Clove (*Syzygium aromaticum*) essential oil is used as an antiseptic against infectious diseases (such as periodontal diseases for example) or in the food industry as an additive or as an antiseptic [44]. The antifungal activity has also been proven in several publications. A MIC at 90% inhibition was demonstrated against *Candida albicans* between 6.2 and 7.5 μL/mL [45].

Dianez et al. determined an ED50 for this oil against several pathogenic strains, for example 447 µg/mL against *Fusarium oxysporum* and 540 µg/mL against *Alternaria brassicae* [24]. 

Finally, the study by Essid et al. determined a MIC between 125 and 250 μg/mL against *C. albicans*, *C. parapsilopsis*, *C. riferii*, *C. tropicalis* and *C. glabrata* [46].

### 2.3. Geraniaceae, Lauraceae and Apiaceae Family 

Highly used in perfumery for its pleasant smell, rose-scented geranium oil (*Pelargonium graveolens*) is also effective against some pathogenic microorganisms and has an antioxidant activity [47]. The effectiveness of this essential oil has repeatedly been shown against various pathogenic fungi. For example, the most frequent MIC determined for more than 100 strains of *C. albicans* and 70 strains of *C. glabrata* was 0.16% *v*/*v*, [48]. Essid et al. also studied the antifungal effect of this oil against *Candida* sp. with a MIC of 500 to 1000 μg/mL against *C. albicans*, 250 μg/mL against *C. tropicalis*, and 500 μg/mL against *C. parapsilopsis*, *C. glabrata* and *C. riferii* [46]. 

The first known use of cinnamon is in food, and this essential oil is used, for example, to flavor chewing gum or toothpaste. This essential oil is being studied for its analgesic, anti-tuberculosis, antioxidant, anti-hyperglycemic and antifungal activity. In addition, some of its compounds could have anti-cancer activities [49]. A study on the effect of *Cinnamomum verum* essential oil on *Fusarium* strains responsible for keratitis in humans was conducted and showed an efficacy at a concentration between 31.25 and 500 μg/mL, depending on the strain studied [50].

Another study concludes that cinnamon essential oil inhibits *A. flavus* growth with a MIC of 100 ppm and has an antimycotoxigenic action [51,52]. In addition, a study of different species of *Candida*: *C. albicans*, *C. parapsilopsis*, *C. riferii*, *C. tropicalis* and *C. glabrata,* revealed a MIC between 31.25 and 62.5 μg/mL [46].

Well known for its use as a spice, cumin (*Cuminum cyminum*) has also been long used as an antiseptic and disinfectant, particularly in India. [53] At present, it is also being studied for its antifungal activity. The study by Kedia et al. on 1230 fungal isolates from food samples, including various strains of *Aspergillus* spp and *Fusarium* spp in particular, demonstrated an activity against these strains with a concentration of 0.6 μL/mL of cumin essential oil [54]. 

In addition, the effectiveness of this oil has been shown on strains of *C. albicans* responsible for gynecological infections, with a MIC between 3.90 and 11.71 μg/mL [55].

To sum up this non-exhaustive list of essential oils with antifungal effects, the main information has been regrouped in Table 2. A summary table of antifungals used as a control is also available to allow a comparison with essential oils (Table 3).

It is also important to remember that the composition of an essential oil varies significantly according to the part of the plant used for oil extraction or depending on the harvest season of the plant, which implies that different studies on the same oil could identify variable compositions (Table 4). Moreover, the great variability of techniques as well as units used by different authors can make the comparison between studies difficult. 

## 3. Natural Compounds and Mechanism of Action

The composition of an essential oil varies not only from one species to another, but also from one organ to another (root or aerial organs, for example), and according to the season [56]. It can contain tens to hundreds of different compounds. Three or four main compounds represent more than 60% of the mass and determine the biological properties of essential oils [11,19]. The major component and its characteristics are summarized in Table 5.

These volatile organic compounds usually have a low molecular weight. They belong to many chemical classes. The majority of compounds derived from essential oils belongs to the family of terpenes and their metabolic derivatives. This family includes functional derivatives of alcohols (geraniol), ketones (menthone), esters (cedryl acetate) and phenols (thymol). In smaller proportions, they also belong to non-terpenic compounds derived from phenylpropane (such as eugenol) [57]. 

The effects of the main actives derived from essential oils may vary depending on whether they are isolated or included in a mixture. In general, three major effects are demonstrated on fungi, a cytotoxic effect, an anti-biofilm one and a third on mycotoxins [58]. Although the precise mechanism of action of essential oils is not yet elucidated, researchers have highlighted some elements. In general, it would appear that essential oils act on several levels, depending on the concentration of the oil. Repeatedly in studies, the plasma membrane and the cell wall appear to be particularly affected. The first effect on the morphology of fungi can be demonstrated during contact with an essential oil (narrow chambered, wide, short or crooked hyphae) [59]. Various studies also show a loss of membrane integrity and a decrease in the amount of ergosterol (major component of the fungal membrane) as well as an inhibition in wall formation. Essential oils also have an inhibitory action on membrane ATPases and cytokine interactions [60], the mitochondria and the endoplasmic reticulum appear to be important sites in their mechanisms of action. Finally, the expression of a certain number of genes seem to be affected, notably, genes involved in adhesion, growth, dimorphism, sporulation, etc.

Due to the fact that some molecules have still not been closely studied, it is difficult to establish all the mechanisms of action. We can only assume that for antifungal efficacy, they depend on the major components [61]. Other compounds have preliminary data that provide an understanding of their mechanisms of action. Only preliminary data are currently known to explain the mechanisms of action of other compounds.

### 3.1. Thymol

Thymol is a terpenoid with a phenol function. It is a volatile compound found mainly in the essential oil of plants in the *Lamiaceae* family such as oregano or thyme [62]. A study of the effect of thymol on *Fusarium graminearum* shows its action on growth with inhibition of hyphal growth. In addition, a decrease in the production and germination of conidia has been demonstrated. A change in the morphology of the fungus has also been observed. Hyphae undergo a loss of cell shape and irregular shrinkage. At a high concentration, the hyphae collapse and break down. Finally, thymol damages the cytoplasmic membrane, leading to the leakage of electrolytes and possibly lipid peroxidation due to ROS generation induced by the increase in permeability [63].

De Castro et al. demonstrated a mode of action on *Candida* spp that is not related to the cell wall biosynthesis pathway but to the binding of thymol to the ergosterol of the membrane, which could lead to a decrease in membrane permeability [64].

In *S. cerevisae*, thymol seems to act on the biosynthetic pathway of the *EST2* gene (coding for the catalytic component of telomerase in yeasts), resulting in a decrease in *EST2* and therefore an inhibition of telomerase activity. The decrease in telomere length can be as much as 13% to 20% and eventually leads to cell cycle arrest, apoptosis and cell death [65].

### 3.2. Carvacrol

Carvacrol is the isomer of thymol. It is the main compound of oregano or savory essential oil [66].

According to a study by Ahmad et al., carvacrol could act by disruption and depolarization of the plasma membrane by targeting membrane proteins and intracellular drug targets [67].

Another study on *Candida albicans* shows that carvacrol could be an endoplasmic reticulum (ER) stressor. The ER of carvacrol-treated cells is fragmented, leading to the disruption of the organization of the ER and to the unfolded protein response, by activating the genes involved in proteolysis, amino acid metabolism, phospholipid translocation, oxidative stress response and DNA repair mechanism, and by inhibiting genes involved in ribosomes biogenesis, glycosylation, sugar transport, drug export and nuclear import. Finally, it was highlighted in this study that this mechanism of action could be different from that of thymol, although the molecules are close [68]. 

### 3.3. Geraniol

Geraniol is an acyclic monoterpenoid found abundantly in lemongrass and aromatic herb oils [69].

A study of the mechanism of action of geraniol is demonstrated by a sorbitol test (osmotic protector) and shows that it does not inhibit fungal cell wall synthesis in *C. albicans*. In addition, after a study on the permeability of the plasma membrane, Leite found that ergosterol is not a direct target of geraniol [70].

Another study revealed that geraniol may induce an inhibitory effect on the calcineurin pathway which leads to damage of the plasma membrane and the cell wall. On the other hand, its effect does not seem to be related to the production of ROS [71].

### 3.4. Cinnamaldehyde

Cinnamaldehyde is the major constituent of *Cinnamomum zeylanicum* (Lauraceae) essential oil [72]. Cinnamaldehyde induces a decrease in the virulence factor in *C. albicans* (decrease in germ tube formation and in adhesion as well as in phospholipase and protease activity, i.e., in the production of hydrolytic enzymes). These effects could be the consequence of an action of this molecule on the ATPase-dependent efflux mechanism [73].

According to the study by Sun et al., cinnamaldehyde has an effect on spore production and fungal growth in *Aspergillus flavus*. It can also inhibit mycelium formation and aflatoxin production (a carcinogenic mycotoxin for humans and animals) and leads to an irreversible transformation on hyphal morphology, a decrease in cytoplasmic content and mitochondria destruction. The permeability caused by this molecule has consequences on the redox and antioxidant systems, which are essential to regulate certain cell signaling pathways (especially in the growth or production of mycotoxins) [74].

The inhibition of aflatoxin in *Aspergillus flavus* was also shown by Liang et al. via the inhibition by this molecule of five genes in its biosynthesis pathway [75].

All of these effects are summarized in Table 6.

## 4. Synergetic Effects of Essential Oils 

The emergent resistance of different pathogenic fungi to classical antifungals is a real human health problem. It would be interesting to combine already known molecules with some essential oil components to bring about synergism, i.e., to have a common effect superior to that of the molecules alone. In addition, a synergistic effect between two compounds could reduce the dose to be used, and thereby reduce the toxicity for humans and the potential side effects of both compounds

The fractional inhibitory concentration index (FICI) estimates the interaction between two or more compounds in order to determine the combined outcome of several molecules. A value less than 0.5 denotes synergism, a value greater than 4 shows antagonism, and between these two values (0.5–4) the combination is considered as indifferent. This FICI is therefore variable depending on the molecules tested but also the targeted species [76].

Several mechanisms seem to be involved in the synergistic effect of antifungals:the inhibition of different stages in the fungal intracellular pathways that are essential for cell survival,the action of another antifungal agent on the fungal cell membrane,the inhibition of carrier proteins,the simultaneous inhibition of different cell targets [77].

Although studies often focus on the major components, these minority components play various roles, in particular in the texture, color or density of the oil, but also in cellular penetration or its lipophilic or hydrophilic nature, its membrane or wall fixation and its distribution within the cell. This implies that studying the whole oil would be more informative about the antimicrobial effect than studying a single component [16]. Several studies show greater antimicrobial efficacy of the whole essential oil than its main component alone. The combination of the major compound and the other remaining components of the oil could have a synergistic effect or a potentiating influence [78]. On the other hand, some minority compounds may have the effect of decreasing the synergistic capacity of the major compound, as alpha-cymene and delta-terpinene do with thymol [79].

### 4.1. Thymol

The combination of thymol and fluconazole has been shown to have a synergistic effect against *T. rubrum* and *A. fumigatus* with a FICI of 0.156 and 0.187, respectively. It divides the MIC of fluconazole by four and that of thymol by 16 for *A. fumigatus* and 32 for *T. rubrum*.

The use of whole thymus oil associated with fluconazole causes a decrease in the synergistic effect, the FICI increases to 0.250 for the two studied species. According to the author, this decrease in the synergistic effect could be due to the alpha-cymene and the delta-terpinene present in this oil at 22.22% and 26.01% [79].

Another study against different strains of the genus *Candida* also highlighted the synergistic effect of the combination of fluconazole and thymol. Eleven strains of each species (*C. albicans*, *C. krusei* and *C. glabrata*) were tested. A synergistic effect was demonstrated for 80%, 90% and 100% of the strains, respectively. The FICI determined for each strain was between 0.366 and 0.607 for *C. albicans*, 0.367 and 0.482 for *C. glabrata* and 0.375 and 0.563 for *C. krusei* [80].

According to Gao et al., the synergistic effect of thymol and fluconazole may be due to the disruption of the wall and membrane integrity, a change in regulation of the mitogen-activated protein kinase (MAPK) pathway, or a creation of plasma membrane lesions combined with a disruption in ergosterol biosynthesis by fluconazole [63].

Other antifungal combinations have also shown a synergistic effect with thymol, such as itraconazole against *P. insidiosum* (FICI of 0.38) and nystatin against *Candida* (FICI of 0.25) [59,76].

### 4.2. Carvacrol

A study on the effect of itraconazole combined with different components derived from this essential oil against azole-sensitive or azole-resistant *Cryptococcus neoformans* showed a synergistic effect when combined with a low dose of carvacrol (>0.1 mg/mL). In fact, the FICI in this study was 0.375 against azole-resistant strains and 0.25 against azole-sensitive strains [81].

Carvacrol combined with fluconazole also has a synergistic effect against *Candida* spp. Synergism was shown for 34 out of 38 sensitive *Candida* strains tested with a FICI between 0.25 and 1.00 and for 10 out of 11 resistant strains tested with a FICI between 0.29 and 0.75 [82]. Finally, the effect was also shown by combining carvacrol with itraconazole in a *P. insidiosum* study (FICI of 0.30) [83].

### 4.3. Geraniol

A study by Singh et al. shows the synergistic effect of geraniol and fluconazole when used together. For this author, the synergistic effect of these two components on the activity of the *C. albicans* efflux pump is a promising way to treat candidiasis. It also shows that although the FICI is greater than 0.5 for amphotericin B and capsofungin (which means that the combination of the compounds does not show a synergistic effect), an increase in efficacy is still noted for these two antifungals at a MIC of 2.5 μg/mL to 62.5 ng/mL for amphotericin B and 0.97 μg/mL to 62.5 ng/mL for caspofungin [84].

Another author also confirmed the synergistic effect of geraniol and fluconazole against *C. neoformans*, with a decrease in the MIC of geraniol from 31.25 μg/mL to 4.14 μg/mL and of fluconazole from 76 μg/mL to 19 μg/mL [85].

### 4.4. Cinnamaldehyde

An amphotericin B/cinnamaldehyde combination was shown to potentiate the efficacy of treatment, even if the effect is not considered as synergistic (FICI > 0.5 for the combination of these two molecules against *C. albicans*). In fact, one study showed a decrease in the MIC of amphotericin B against *C. albicans* with an essential oil containing more than 90% cinnamaldehyde [86]. Another study showed a synergistic effect of cinnamaldehyde and fluconazole against *T. rubrum* by decreasing the MIC of fluconazole by more than 8-fold, with a FICI of 0.312. On the other hand, this combination is not synergistic against *A. fumigatus* (FICI of 0.625) [79]. Finally, the synergistic effect of cinnamaldehyde combined with fluconazole has also been demonstrated against multi-resistant strains and against biofilms formed by *C. albicans* [67,82].

## 5. Toxicity and Side Effects

Although essential oils are important, they are not devoid of adverse effects even if a lot of essential oil components are GRAS (generally recognized as safe). In fact, 80 different oils are responsible for allergies or dermatitis [18]. They can also cause stomatitis or ototoxicity. The systemic administration of the oil can cause neuro- or hepato-toxicity, kidney irritation and changes in the intestinal mucosa. Some of them, such as citrus oil, can also be phototoxic [87]. These effects are clearly related to the dosage, the composition of the oil (variable depending on the season or plant used), the method of administration, the health status of the person using it or the additives in the oils [30]. Oxidation due to oil conservation can also play a role in toxicity [88].

Tea tree oil, like every essential oil, can also cause irritation and allergies. They seem to occur especially in case of a poorly preserved or old essential oil [39]. Another example is cinnamon oil, known to cause contact allergies and can lead to mild pain, pruritus or a burning sensation [89]. Several side effects have already been highlighted after the use of mint essential oil. For example, it can cause nausea and flushing in addition to allergic reactions already attributed to essential oils. These effects could be due to drug interactions caused by the oil. In fact, peppermint is known to interfere with cytochrome P450, which plays an important role in drug metabolism [90].

An in vitro study on fibroblasts made it possible to demonstrate a toxic effect of clove oil at a concentration of 0.03% (1.8 μM). This toxicity was mainly due to eugenol, a major clove oil component, leading to plasmin membrane damage or to an apoptotic effect [91].

Other damages have been noted at a high concentration of the oil in in vitro studies. For example, they can cause ultrastructural changes (nucleolar segregation, lipid degeneration, damaged mitochondria) [92].

## 6. Resistance

Little data exists on the resistance induced by essential oils, and specifically on the fungi responsible for infections. Most of the studies concern pathogenic bacteria. Some bacteria showed resistance against essential oils, for example a study on *Pseudomonas aeruginosa* showed an increase in the tolerance against the compounds of tea tree oil during contact with a sub-inhibitory concentration [93].

It has also been shown that the presence of a sub-inhibitory concentration of essential oil can affect the sensitivity of bacteria to antibiotics [94]. For example, the cultivation of *P. mirabilis* in oregano essential oil increased the MIC to ampicillin 8-fold. This phenomenon depends on the species and oil studied. In fact, exposure of *P. aeruginosa* to oregano or cinnamon oil does not change the MIC for all antibiotics tested [95].

The resistance induced by essential oils is still very poorly understood and requires further studies. It is currently considered to be generally weak, reversible and dependent on the species studied as well as the oil studied. The possibility of an innate resistance of certain species should also be taken into consideration [88].

## 7. Example of Patents of Natural Compounds with Antifungal Properties

Because of their numerous antimicrobial effects, it is not surprising to find many projects on this subject in the literature. There are also a number of patents pending to advance the quest for an effective new antifungal drug and some of which are described below. Patents can be seen on the website of the World Intellectual Property Organization.

On 31 August 2006 a patent proposed by Weiqi and Wenwen for an essential oil gel coat to fight onychomycosis was granted. The product has a synergistic interaction between its compounds that allows an effective fungicidal action.

Another patent based on natural product is a bath salt (containing *Melissa officinalis* essential oil) capable of removing dirt from human skin while having antifungal, anti-inflammatory and skin oxidation resisting action. This product was created by Meirong and patented on 1 March 2019.

The proposed patents concerning essential oils not only concern the medical field. On 14 March 2019, a project of Satyawati, Abhishek, Saurabh and Naik on a microemulsion composed of two essential oils was published. The goal was to facilitate the overall antifungal efficacy of the microemulsion but also to maintain its power against the fungus *F. oxysporum*, which induces the wilt during storage.

Finally, Ramirez and Sanchez have created a degradable film for fruit packaging (accepted 10 January 2019). This film contains a polymer matrix that incorporates an antimicrobial active ingredient of an essential oil and a degrading agent.

## 8. Conclusions

Essential oils are already widely used in various fields such as cosmetology or the food industry. Numerous studies have also highlighted the action of these oils on various physiological effects in humans. In addition, their antimicrobial effects have also been recognized.

The antifungal effects of these oils, alone or in combination with pre-existing antifungals, would be a very interesting solution for improving therapeutic failures related to emergence of resistant strains and an increase in immunosuppressive treatments in medicine.

Although the studies are promising, we must not forget to consider the toxicity of these products. In fact, essential oils are already known to pose a high risk of allergy and irritation. Therefore, it is important to continue studies to accurately determine all the risks they might cause. In addition, it should be noted that there are many essential oils and each one is different, with a panel of various compounds. Therefore, it is necessary to specifically study one essential oil in particular against a given pathogenic species, or even to study the effect of the different compounds alone. There is still a lot of work to be done before we can actually include essential oils in current medical treatments, but they remain a good candidate for future treatments and to compensate for therapeutic failures. Moreover, knowledge of the mechanism of action of a compound derived from essential oil could tell us with which molecule they are combined or prevent possible resistance.

## Figures and Tables

**Table 1 molecules-24-03713-t001:** Invasive fungal infections (IFI) data in the world [3].

IFI	Localization	Cases/Year	Mortality Rate
Invasive Aspergillosis	Worldwide	>200,000	30%–95%
Invasive Candidiasis	Worldwide	>400,000	46%–75%
Cryptococcosis	Worldwide	>1,000,000	20%–70%
Mucormycosis	Worldwide	>10,000	30%–90%

**Table 2 molecules-24-03713-t002:** Converted results of the various publications cited in order to homogenize the results. It can be considered that in general, the concentration of a pure essential oil is 1 g/mL [30]. With no further precision, the values corresponding to an 100% inhibition.

Essential Oil	Major Compounds	Pathogens Tested	MIC/Concentration Used in the Studies	Converted Values (µg/mL)	Number of Strains Tested	% of the Major Compound (When Presided)	References
*Thymus vulgaris*	Thymol Carvacrol p-Cymene	*Candida albicans*	62 µg/mL		1	Thymol 60.8% Carvacrol 2.88% p-Cymene 15.4%	[67]
		*Candida tropicalis*					
		*Fusarium* sp	ED50 71 µg/mL		1	Thymol 0.2% Carvacrol 81.5%	[21]
		*Aspergillus* sp	9.85 µg/mL		44	Thymol 33% Carvacrol 3.9%	[22]
		*Penicillium* sp	19.17 µg/mL		18		
		*Cladosporum* sp	15.20 µg/mL		6		
		*Botrytis cinerea*	-		1	-	[23]
		*Alternaria brassicae*	ED50 67.7% *v*/*v*	ED50 677 µg/mL	1	-	[24]
		*Fusarium oxysporum*	ED50 36.3% *v*/*v*	ED50 363 µg/mL	1		
*Thymus pulegioides*		*Fusarium graminearum*	105–108 µg/mL		1	-	[25]
		*Aspergillus* sp	0.16–0.64 µL/mL	160-–40 µg/mL	9	Thymol 26% Carvacrol 21%	[20]
		*Dermatophytes*			5		
		*Candida* sp			11		
*Maleleuca alternifolia*	Terpinen-4-ol	*Aspergillus* sp	0.016%–0.12% *v*/*v*	1.6–200 µg/mL		Terpinen-4-ol 40.1%	[39]
		*Candida* sp	0.03%–8% *v*/*v*	3–800 µg/mL			
		*Candida albicans*	1.95 mg/mL		1	-	[40]
*Origanum vulgare*	Thymol Carvacrol Sabinene Linalool	*Botrytis cinerea*	ED50 50 µg/mL		1	Thymol 63.7% Carvacrol 8.6%	[21]
		*Fusarium* sp			1		
		*C. albicans*	1.48–1.75 mg/mL		1	Carvacrol 39.08%–49.03% Sabinene 1 9.81%–25.11%	[27]
		*A. niger*	2.75–2.85 mg/mL		1		
		*C. glabrata*	0.5–1100 µg/mL		16	Thymol 25.1% Linalool 42%	[28]
		*A. flavus*	400 ppm	3.6 ug/mL	1		[51]
*Mentha piperita*	Linalool Menthol Piperitone	*Candida albicans*	1 µL/mL	1 mg/mL	1	Piperitone 38% Piperitenone 33%	[29]
		*Aspergillus niger*	0.25 µL/m	250 mg/mL	1		
		*Candida* sp	800 µg/mL			-	[30]
		*Aspergillus* sp	222 µg/mL				
*Mentha pulegium*	Pulegone	*Candida albicans*	500 ppm	44.5 µg/mL	1	Menthol 37.88%	[32]
		*Candida* sp	400–7000 µg/mL			-	[30]
		*Dermatophyte*	800–3500 µg/mL				
		*Aspergillus* sp	400–3500 µg/mL				
*Lavendula angustifolia*	Linalool Linalyl acetate	*b.cinerea*	ED50 223 µg/mL		1	Linalool 25.5% Linalyl acetate 17.7%	[21]
		*Fusarium* sp	520 µg/mL		1		
		*F.oxysporum*	ED50 372 µL/mL	37.2 mg/mL	1	-	[24]
		*C. albicans*	1/40 of pure solution of essential oil		20	-	[33]
		*C. albicans*	5000 ppm	445 µg/mL	50	Linalool 24.7% Linalyl acetate 31.1%	[34]
*Rosmarinus officinialis*	1,8-Cineole Camphor α-pinene	*B.cinierea*	ED50 600 µg/mL		1	Eucalyptol 31.5%	[21]
		*Fusarium* sp	660 µg/mL		1		
		*C. albicans*	MIC 80% 24–31 µg/mL		11	1,8-Cineole 31.5%	[36]
		*C. albicans*	0.78 mg/mL		1	1,8-Cineole 52.2% Camphor 15.2% α-pinene 12.4%	[37]
*Pelargonium graveolens*	(Z)-geraniol Citronellol	*C. albicans*	0.16% *v*/*v*	1.6 mg/mL	47	Citronellol 11.94%	[48]
		*C. glabrata*			20		
		*C. albicans*	500–1000 µg/mL		5	Citronellol 27.23%	[46]
		*C. tropicalis*	250 µg/mL		1		
		*C. parasilopsisi*	500 µg/mL		1		
		*C. glabrata*	500 µg/mL		2		
		*C. riferi*	500 µg/mL		1		
*Eucalyptus citriodora*	Citronellol Citronellal	*C. albicans*	318 µg/mL		1	-	[42]
*Eucalyptus camaldulensis*	p-cymene 1,8-Cineole	*P. funicuarum*	0.15 mg/mL			-	[43]
		*A. niger*	0.47 mg/mL				
		*A. flavus*	0.43 mg/mL				
*Cinnamomum verum*	Cinnamaldehyde	*Fusarium*	31.25–500 µg/mL		18	Cinnamaldehyde 93.1%	[50]
		*A. flavus*	100 ppm	8.9 µg/mL	1	-	[52]
		*C. albicans*	31.25–62.5 µg/mL		5	Cinnamaldehyde 82.09%	[46]
		*C. parasilopsis*			1		
		*C. riferii*			1		
		*C. tropicalis*			1		
		*C. glabrata*			2		
*Cuminum cyminum*	Cuminaldehyde Cymene γ-terpinene 1,8-cineole	*Fusarium* sp	0.6 µL/mL	600 µg/mL	1230	Cymene 47.8% Cuminaldehyde 14.92% γ-terpinene 19.36%	[54]
		*Aspergillus* sp					
		*C. albicans*	3.90–11.71 µg/mL		20	1,8-cineole 21.07%	[55]
*Sysygium aromaticum*	1,8-cineole Eugenol	*C. albicans*	MIC 50% 6.2–7.5 µL/mL	6.2–7.5 mg/mL	38	Eugenol 76.84%	[45]
		*A. brassicae*	ED50 54% *v*/*v*	540 µg/mL	1	Eugenol 86.38%	[24]
		*F.oxysporum*	ED50 44.7% *v*/*v*	447 µg/mL	1		
		*C. albicans*	125–250 µg/mL		5	Eugenol 90.43%	[46]
		*C. parapsilopsis*			1		
		*C. riferii*			1		
		*C. tropicalis*			1		
		*C. glabrata*			2		

MIC: minimal inhibitory concentration. ED50: effective dose 50.

**Table 3 molecules-24-03713-t003:** Comparison between essential oils MIC and those of reference antifungals (when tested).

Essential Oil	Pathogens Tested	MIC/Concentration Used in the Studies (Converted)	Atf Tested	MIC	References
*Thymus vulgaris*	*Candida albicans*	62 µg/mL	Amphotericin B	0.001 mg/mL	[67]
	*Candida tropicalis*		Amphotericin B	0.001 mg/mL	
*Thymus pulegioides*	*Aspergillus* sp	160–640 µg/mL	Amphotericin B	2–8 µg/mL	[20]
	*Dermatophytes*		Fluconazole	16–168 µg/mL	
	*Candida* sp		Fluconazole	1–168 µg/mL	
*Origanum vulgare*	*C. albicans*	1.48–1.75 mg/mL	Amphotericin B	100 µg/mL	[27]
	*A. niger*	2.75–2.85 mg/mL	Amphotericin B	100 µg/mL	
*Mentha piperita*	*Candida albicans*	1 mg/mL	Amphotericin B	1 µL/mL	[29]
	*Aspergillus niger*	250 mg/mL	Amphotericin B	0.25 µL/mL	
*Pelargonium graveolens* *Cinnamomum verum* *Sysygium aromaticum*	*C. albicans*	500–1000 µg/mL	Amphotericin B Fluconazole	0.5–2 (µg/mL) 62.5–1000 (µg/mL)	[46]
	*C. tropicalis*	250 µg/mL	Amphotericin B Fluconazole	2 (µg/mL) 1000 (µg/mL)	
	*C. parasilopsisi*	500 µg/mL	Amphotericin B Fluconazole	2 (µg/mL) 7.81 (µg/mL)	
	*C. glabrata*	500 µg/mL	Amphotericin B Fluconazole	2 (µg/mL) 15.62–31.25 (µg/mL)	
	*C. riferi*	500 µg/mL	Amphotericin B Fluconazole	2 (ug/mL) 1000 (ug/mL)	
*Cinnamomum verum*	*Fusarium* sp	31.25–500 µg/mL	Natamycin	2–256 ug/mL	[50]
*Cuminum cyminum*	*C. albicans*	3.90–11.71 µg/mL	Fluconazole	3.24–54 ug/mL	[55]

**Table 4 molecules-24-03713-t004:** Part of the plants used for extracting essential oils.

Essential Oil	Part of the Plant	Familly	References
Thyme	Leaves Aerial part	*Lamiaceae*	[20,21]
Tea Tree	Leaves	*Myrtaceae*	[21]
Origano	Leaves	*Lamiaceae*	[21,27]
Mentha	Flowering aerial part	*Lamiaceae*	[29]
Lavander	Aerial part	*Lamiaceae*	[33]
Rosmarin	Aerial part Leaves	*Lamiaceae*	[21,36,37]
Geranium	Aerial Part	*Geraniaceae*	[46]
Eucalyptus	Leaves	*Myrtaceae*	[42,43]
Cinnamon	Leaves Bark	*Lauraceae*	[46,51]
Cumin	Seeds Aerial part	*Apiaceae*	[54,55]
Clove	Leaves	*Myrtaceae*	[45,46]

**Table 5 molecules-24-03713-t005:** Characteristics of the major compounds of essential oils (PubChem).

Compounds	Molecular Formula	Essential Oil	Structure
Thymol 2-Isopropyl-5-methylphenol	C_10_H_14_O	Thyme	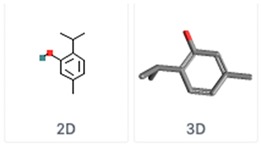
Carvacrol 5-Isopropyl-2-methylphenol	C_10_H_14_O	Oregano Savory	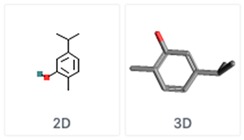
Geraniol Geranyl alcohol	C_10_H_18_O	Lemongrass	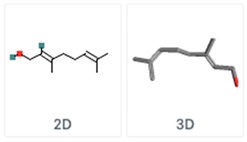
Cinnamaldehyde trans-Cinnamaldehyde	C_9_H_8_O	Cinnamon	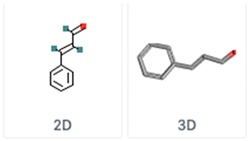

**Table 6 molecules-24-03713-t006:** Summary of the effects of the compounds derived from essential oils.

Molecules	Mechanism of Action	Species Tested	Reference
Thymol	Ergosterol binding: plasma membrane permeability	*Candida* spp	[64]
	Inhibition of hyphal growth, conidia production and germination Electrolytes leakage Lipid peroxidation	*F. graminearum*	[63]
	Telomerase activity inhibition: cell death, stop of the cellular cylce, apopotoe	*S. cerevisae*	[65]
Carvacrol	Targeting plasma membrane protein and intracellular target: disruption and depolarization of the plasma membrane Endoplasmic reticulum disruption: unfold protein response	*C. albicans/C. tropicalis* *C. albicans*	[67] [68]
Geraniol	Inhibition of the calcineurin pathway: plasma membrane and cell wall damage, ROS production	*C. albicans*	[71]
Cinnamaldehyde	Decrease of the virulence factors Effect on spore production, fungal growth and aflatoxine	*C. albicans* *A. flavus*	[73] [74,75]

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
