# Peer review of "Essential Oils and Their Natural Active Compounds Presenting Antifungal Properties"

_molecules, 2019, doi:10.3390/molecules24203713_

Round 1

Reviewer 1 Report

The review manuscript entitled “Natural active compounds isolated from essential oils 2 and their antifungal properties” is presenting important aspects from recent literature dealing with antifungal activity of essential oils. The manuscript is quite well written, however, there are some important aspects that have to be clarified in order to be publishable in Molecules, as indicated bellow:

Introduction In the first paragraphs, most of the data dealing with public health problems are from France. I recommend the authors to add information regarding worldwide situation or Europe at least (with reference). Information regarding incidents of phytopathogenic fungi in plants from crops and the importance in economic loss, are also necessary (with reference) since the authors discuss the influence of the essential oil on phytopathogenic fungi as well (Botrytis cinerera, Fusarium sp., Alternaria brassicae etc.) Essential oils The word “antifungal” and “antioxidant” (and others similar) should be consistent through the manuscript (ex: in page 5 “anti-fungal”, “anti-oxidant” were used). The concentration 1g/mL mention in table 1 is confusing (the cited reference does not explain it). Why do the authors have to mention it? What does it mean? 1 mL of essential oil cannot have 1 g since the density is always lower than water’s. If the essential oil is pure, you cannot talk about concentration. Concentration is used for solutions. Each essential oil will have different g/mL, depending upon the density. Why do the authors bold the first fungus line (Candida albicans)? The information about “a delocalized electron” in carvacrol, page 8 bottom, it has no importance and even non-scientific. The cited reference (62) is not the initial mention of the effect, it is rather it is another from Appl Environ Microbiol. 2002 Apr; 68(4): 1561–1568. However, the effect (pH effect) is arguable not relevant.

Overall

There are too many small paragraphs with lots of new tabs which make the reading a bit fractured. Other already existent review regarding essential oils with antifungal activity are not mentioned. What bring new this manuscript compared to the others already available? The partition of the review in only two parts (1. Introduction and 2. Essential oils) is not very welcome. Since the authors discuss other aspects, I recommend at least three or four main parts.

Author Response

Maurine D’Agostino

PhD student at Université de Lorraine

EA SIMPA

54511 Vandoeuvre les Nancy

[email protected]

Nancy, October 3th, 2019

Dear Editor and Reviewers,

Thank you very much for your comments and suggestions. We have been attentive to these comments. Please, find detailed responses to the points you raised. Changes are marked in colours in the text.

Briefly, as suggested by the reviewers, we provided some clarifications for a better understanding of our work.

Your sincerely,

Maurine D’agostino

REVIEWER 1

« Overall, this review is well presented and organized. Instead of just focusing on papers describing the mechanisms of action of individual compounds on essential oils, I advise the authors to include studies on the mechanism of action of whole essential oils. »

Thank you for your comment, I have added some more global data on invasive fungal infection, especially Table 1 (Invasive fungal infections in the world)

« The information provided in the paper should be aided by more figures/tables were the information can be more clearly read. A review paper with just one scarce table is not suited for this type of papers.»

After your comment I chose to add several figures :

Table 1. Invasive fungal infections in the world

Table 3. Comparison with reference antifungal (when tested)

Table 4. Part of the plants used for making essential oil

Table 5. Details of the major compounds

« Correct CMI by MIC »

 The correction has been made line 152 and in the table 2.

« Some important references on the topic are missing: 

Antifungal activity of Coriandrum sativum essential oil, its mode of action against Candida species and potential synergism with amphotericin B Combined analytical and microbiological tools to study the effect on Aspergillus flavus of cinnamon essential oil contained in food packaging

ANTIFUNGAL ACTIVITY OF THE ESSENTIAL OIL OF CINNAMON (CINNAMOMUM ZEYLANICUM), OREGANO (ORIGANUM VULGARE) AND LAURAMIDE ARGINE ETHYL ESTER (LAE) AGAINST THE MOLD ASPERGILLUS FLAVUS CECT 2949

 Diminution of aflatoxin B1 production caused by an active packaging containing cinnamon essential oil. »

The bibliography has been incremented with these indexed references 2, 49 and 50.

Reviewer 2 Report

Overall, this review is well presented and organized. Instead of just focusing on papers describing the mechanisms of action of individual compounds on essential oils, I advise the authors to include studies on the mechanism of action of whole essential oils. The information provided in the paper should be aided by more figures/tables were the information can be more clearly read. A review paper with just one scarce table is not suited for this type of papers.

Correct CMI by MIC

Some important references on the topic are missing:

Antifungal activity of Coriandrum sativum essential oil, its mode of action against Candida species and potential synergism with amphotericin B Combined analytical and microbiological tools to study the effect on Aspergillus flavus of cinnamon essential oil contained in food packaging ANTIFUNGAL ACTIVITY OF THE ESSENTIAL OIL OF CINNAMON (CINNAMOMUM ZEYLANICUM), OREGANO (ORIGANUM VULGARE) AND LAURAMIDE ARGINE ETHYL ESTER (LAE) AGAINST THE MOLD ASPERGILLUS FLAVUS CECT 2949 Diminution of aflatoxin B1 production caused by an active packaging containing cinnamon essential oil

Author Response

Maurine D’Agostino

PhD student at Université de Lorraine

EA SIMPA

54511 Vandoeuvre les Nancy

[email protected]

Nancy, October 3th, 2019

Dear Editor and Reviewers,

Thank you very much for your comments and suggestions. We have been attentive to these comments. Please, find detailed responses to the points you raised. Changes are marked in colours in the text.

Briefly, as suggested by the reviewers, we provided some clarifications for a better understanding of our work.

Your sincerely,

Maurine D’agostino

REVIEWER 2

«Introduction In the first paragraphs, most of the data dealing with public health problems are from France. I recommend the authors to add information regarding worldwide situation or Europe at least (with reference). »

Thank you for your comment, following your advice, I added the table 1 : « Invasive fungal infections in the world » in the introduction.

Information regarding incidents of phytopathogenic fungi in plants from crops and the importance in economic loss, are also necessary (with reference) since the authors discuss the influence of the essential oil on phytopathogenic fungi as well (Botrytis cinerera, Fusarium sp., Alternaria brassicae etc.)

A short paragraph on “Crop disease and fungal infection” has been added Line 53 to 64 to highlight the importance of phytopathogenic fungal infections.

Essential oils: The word “antifungal” and “antioxidant” (and others similar) should be consistent through the manuscript (ex: in page 5 “anti-fungal”, “anti-oxidant” were used).

The correction was made line 208 « antioxydant » and line 209 « antifungal ».

The concentration 1g/mL mention in table 1 is confusing (the cited reference does not explain it). Why do the authors have to mention it? What does it mean? 1 mL of essential oil cannot have 1 g since the density is always lower than water’s. If the essential oil is pure, you cannot talk about concentration. Concentration is used for solutions. Each essential oil will have different g/mL, depending upon the density.

Sorry for the inconvenience, an error was made in the added reference in the table 2. The good one was Stringaro et al., 2018

 « Therefore, comparing the antifungal activity of different EOs is very difficult, as different methods and ways of expressing concentrations have been used. Sometimes, some authors have reported values of MIC as percent v/v, μL/mL, or μg/mL. It is more important to highlight that the essential oil weight is about 1 g/mL. When the values of MIC were expressed in μL/mL, a few μL correspond to milligrams of essential oil. The same is true when the MIC values were expressed in percentages. »

« Why do the authors bold the first fungus line (Candida albicans)? »

The error has been corrected.

« The information about “a delocalized electron” in carvacrol, page 8 bottom, it has no importance and even non-scientific. »

The information has been removed following your note.

« The cited reference (62) is not the initial mention of the effect, it is rather it is another from Appl Environ Microbiol. 2002 Apr; 68(4): 1561–1568. However, the effect (pH effect) is arguable not relevant. »

The information has been removed following your note.

« There are too many small paragraphs with lots of new tabs which make the reading a bit fractured. »

A reshuffle of  the sections into 3 sub-sections by plant family was made.

« Other already existent review regarding essential oils with antifungal activity are not mentioned. What bring new this manuscript compared to the others already available? »

Like all revues, this work has been done to bring knowledge on a specific topic, bringing together the results of different writers. This work updates the data to date in a single document with a summary table to quickly compare the different studies available to date.

« The partition of the review in only two parts (1. Introduction and 2. Essential oils) is not very welcome. Since the authors discuss other aspects, I recommend at least three or four main parts. »

Thank you for your comment, several sub parts have been created to answer your order :

1.Introduction ; 2.The main essential oils with antifungal activity . 3. Mechanism of action ; 4.Synergetic effects of essential oils ; 5.Toxicity and side effects ; 6.Resistance ; 7. Example of patents of natural compounds with antifungal properties ; 8. Conclusion

Reviewer 3 Report

Dear,

All review articles are very important for increasing scientific and technological knowledge. However, the authors did not illustrate Figures of 2D or 3D structures of major compounds.

Another point to note is the inclusion (creation of a topic) of natural compounds that have been patented or are being patented with antifungal properties.

The authors only cite compounds, but at no time attempt to clarify possible mechanism of action in vitro, in vivo or in silico to readers. Likewise, the authors could enter data to search or new compounds design with potential antifungal activity. For example, see the article Computational Investigation of Antifungal Compounds Using Molecular Modeling and Prediction of ADME / Tox Properties - https://doi.org/10.1166/jctn.2015.4260.

In general the article needs to be improved and focused on the solution of the mechanism of action of the isolated or major compounds of the species cited in the text, as the journal Molecules aims to clarify possible mechanisms of action, which may be in silico or in vitro or in vivo. .

Author Response

Maurine D’Agostino

PhD student at Université de Lorraine

EA SIMPA

54511 Vandoeuvre les Nancy

[email protected]

Nancy, October 3th, 2019

Dear Editor and Reviewers,

Thank you very much for your comments and suggestions. We have been attentive to these comments. Please, find detailed responses to the points you raised. Changes are marked in colours in the text.

Briefly, as suggested by the reviewers, we provided some clarifications for a better understanding of our work.

Your sincerely,

Maurine D’agostino

REVIEWER 3

« All review articles are very important for increasing scientific and technological knowledge. However, the authors did not illustrate Figures of 2D or 3D structures of major compounds. »

After your comment I chose to add a table with 2D structures of major compounds (table 5 section 4).

« Another point to note is the inclusion (creation of a topic) of natural compounds that have been patented or are being patented with antifungal properties. »

 Thank you for your comment, I have added a topic about patents of essential oil and antifungal activities (Section 7).

« The authors only cite compounds, but at no time attempt to clarify possible mechanism of action in vitro, in vivo or in silico to readers. Likewise, the authors could enter data to search or new compounds design with potential antifungal activity. For example, see the article Computational Investigation of Antifungal Compounds Using Molecular Modeling and Prediction of ADME / Tox Properties - https://doi.org/10.1166/jctn.2015.4260. »

« In general the article needs to be improved and focused on the solution of the mechanism of action of the isolated or major compounds of the species cited in the text, as the journal Molecules aims to clarify possible mechanisms of action, which may be in silico or in vitro or in vivo. »

Today, little data is available on the mechanism of action of natural substances with antifungal activities. This demonstrates the value of basic research work in this area. In the introduction of the "mechanism of action" section, we have described the different principles of action of an essential oil. Then, for the most studied molecules, we presented the observations and the first mechanistic hypotheses described by the teams working on these natural substances.

Reviewer 4 Report

Minor revisions should be made, and the manuscript should be completed and/or modified taking into account the suggestions from the attached file.

Author Response

Maurine D’Agostino

PhD student at Université de Lorraine

EA SIMPA

54511 Vandoeuvre les Nancy

[email protected]

Nancy, October 3th, 2019

Dear Editor and Reviewers,

Thank you very much for your comments and suggestions. We have been attentive to these comments. Please, find detailed responses to the points you raised. Changes are marked in colours in the text.

Briefly, as suggested by the reviewers, we provided some clarifications for a better understanding of our work.

Your sincerely,

Maurine D’agostino

REVIEWER 4

« The authors should prepare their manuscript according to Instructions for authors: „In the text, reference numbers should be placed in square brackets [ ], and placed before the punctuation; for example [1], [1–3] or [1,3]” »

The corrections were made lines 36, 41, 91, 254, 262, 377, 408 according to your comment.

« The authors should present either the name(s) of the species, either spp. after the genus (Scedosporium, Fusarium) -line 38. The same for line 99, 100-101, 113-114, 122, 195-196, 359, etc. Also, they should correct „ muccorales” (line 39). »

Thank you for your corrections, the corrections were made lines 38, 120, 121,122,134, 135, 143, 220, 383 « spp » and Line 39 « mucorales ».

« The authors should rephrase the following: „These therapeutic activities are carried out by different classes of molecules: terpenoid and phenylpropanoid.” (lines 73-74) »

The sentence has been changed line 90-91 : « Many classes of molecules contained in essential oil like terpenoid and phenylpropanoid give them this therapeutic activities »

« The authors should better explain why they choose to present the eleven essential oils. »

Following your comment, an explanation was made lines 112-113: « The essential oils presented in this review were chosen for their proven antifungal effect and their frequent use. Many others exist elsewhere, but we chose to study a limited number so as not to overload this presentation.».

« The authors should use the italic style for Latin names of plants (line 93, 115, 124, etc) »

The titles of the sub-section has been removed after changing the different sub sections.

«  Also, the authors should present which vegetal product was uased in order to obtain each essential oil, since their composition is different from one plant organ to another (in subsections 2.1.1, 2.1.2, 2.1.3, etc ) »

An other table was made to respond to your comment : « Table 4. Part of the plants used for making essential oil ».

« The authors should rephrase the following: „The use of thyme essential oil is well known in fungal infections that affect humans, with its high concentration of thymol and carvacrol, it has a broad spectrum of action on most pathogenic fungi” »

The sentence has been changed line 116-117 : « Thyme essential oil is already known to be effective against fungi infecting humans. Its antifungal activity is due to its high concentration of thymol and carvacrol. »

« The authors are advised to use et al. after the name of the first author, since there are several authors of each paper (ex. Ahmad et al., Daferera et al., etc) »

The correction has been made : Line 118 « Ahmad et al., » ; Line 119 « Daferera et al., » ; line 121 « Klaric et al., » ; line 133 « Pinto et al., » ; line 141 « Carson et al., » ; line 150 «  Daferera et al., » ; line 152 «  Santoyo et al ., » ; line 153 « Khosravi et al., » ; line 160 « Mahboudi et al., » ; line 163 « Stringaro et al., » ; line 175 « Daferera et al., » ; line 176 « Diànez et al., » ; line 178 « Behmanesh et al., » ; line 185 « Daferera et al ., » ; line 188 « Oliveira et al. » ; line 194 « Essid et al., » ; line 203 « Elansany et al., » ; line 219 « Kedia et al. » ; line 229 « Dianez et al., » ; line 231« Essid et al., » ; line 289 « De Castro et al., » ; line 299 «Ahmad et al., » ; line 325 « Sun et al., » ; line 332 « Liang et al., » ; Line 372 « Gao et al., » ; line 389 « Singh et al., » ;

« The authors should also present the reference drugs used for the evaluation of antifungal effects, as well as the obtained MIC/MFCs, in order to evaluate the obtained results. »

Another table was made to respond to your comment : « Table 3. Comparison with reference antifungal (when tested) ».

« The authors should better explain why the large interval (0.5-1,100 μg/ml) was obtained – lines 137 »

A better explanation was made line 155 : « although the difference between MICs for the same species is important, it is not surprising. Indeed, each strains can respond differently according to their own sensitivity. »

« The authors should rephrase the following: „For example, Behmanesh showed significant activity by diluting a pure solution of essential oil of lavender [33] to one fortieth, Moreover concentrations between 0.125 and 2% v/v could completely inhibit its growth [34].” (lines 152-154) »

The sentence has been changed lines 177-180 : « For example, a study by Behmanesh et al. shows a significant antifungal effect of lavender essential oil after diluting 40 times the pure solution. Finally, a concentration of between 0.125 and 2% v / v completely inhibits its growth. »

«  The titles of some subchapters should be changed, for example „2.1.8. Eucalyptus (Eucalyptus citriodora) essential oil”, since there are two Eucalyptus species presented. (line 171). The same for 2.1.4. »

The titles of the sub-section was removed to restructure this section.

« The authors should rephrase the following: „In addition, the effect of this essential oil on C. albicans strains responsible for gynecological infection was demonstrated., highlighting a MIC between 3.90 and 11.71 μg/ml” »

The sentence has been changed line 222-223 : « In addition, the effectiveness of this oil has been shown on strains of C. albicans responsible for gynecological infection, with a MIC between 3.90 and 11.71 μg / ml »

« The authors are advised to change „different compositions” »

The word was changed line 237 : « Variable composition »

« In Table 1, the authors should also present the percentage of each major compound identified in each essential oil »

Following your advice, a column was created in the table 2 : « % of the major compounds ».

« The authors are advised to change the title of subchapter 2.2, since the compounds are presented in 2.2.1, 2.2.2, etc. »

The title has been changed « Antifungal capacity and mechanism of action »

« The authors should rephrase the following: „Another T. rubrum study in which cinnamaldehyde and fluconazole were combined showed a more than 8-fold decrease in the MIC of fluconazole when cinnamaldehyde was added, with a FICI of 0.312.” »

The sentence has been changed line 403-405 : « Another study showed a synergistic effect of cinnamaldehyde and fluconazole against T.rubrum by decreasing the MIC of fluconazole by more than 8-fold, with a FICI of 0.312. »

« The authors should rephrase the following: „Moreover, knowledge of the mechanism of action of a compound in an essential oil could provide us with information on when it can be used with another molecule or to pr"edict the possible resistances” »

The sentence has been changed line 465-466 : « Moreover, knowledge of the mechanism of action of a compound derived from essential oil could tell us with which molecule it could be combined or prevent possible resistance »

Round 2

Reviewer 1 Report

The manuscript is improved and could be accepted for publication.

Just minor text editing and the chemical structure of the major compounds (Table 5) could be improved for their quality.

Author Response

Maurine D’Agostino

PhD student at Université de Lorraine

EA SIMPA

54511 Vandoeuvre les Nancy

[email protected]

Nancy, October 10th, 2019

Dear Editor and Reviewers,

Thank you very much for your comments and suggestions. We have been attentive to these comments. Please, find detailed responses to the points you raised. Changes are marked in colours in the text.

Briefly, as suggested by the reviewers, we provided some clarifications for a better understanding of our work.

Your sincerely,

Maurine D’agostino

Reviewer 1

“The manuscript is improved and could be accepted for publication.

Just minor text editing and the chemical structure of the major compounds (Table 5) could be improved for their quality.”

The table 5 has been improve according to your advice

Reviewer 2 Report

The manuscript has been substantially improved. However, there are still some issues regarding:

the correct use of English language and misspelled words the main focus of the paper: the title says "natural active compounds isolated from EO" and instead of focusing on the compounds, the authors focus on the EOs. Therefore, the paper should be rewritten to focus on the main natural compounds from EOs as it is the main topic of the paper There are Tables/Figures lacking in the section related to the mechanism of action Some tables are not cited in the text The tables provided are quite confusing and should be edited in a different form The authors focus too much on French reality, while the paper should give a worldwide broad view on the topic 

Author Response

Maurine D’Agostino

PhD student at Université de Lorraine

EA SIMPA

54511 Vandoeuvre les Nancy

[email protected]

Nancy, October 10th, 2019

Dear Editor and Reviewers,

Thank you very much for your comments and suggestions. We have been attentive to these comments. Please, find detailed responses to the points you raised. Changes are marked in colours in the text.

Briefly, as suggested by the reviewers, we provided some clarifications for a better understanding of our work.

Your sincerely,

Maurine D’agostino

Reviewer 2 

The manuscript has been substantially improved. However, there are still some issues regarding:

“the correct use of English language and misspelled words the main focus of the paper: The title says "natural active compounds isolated from EO" and instead of focusing on the compounds, the authors focus on the EOs. Therefore, the paper should be rewritten to focus on the main natural compounds from EOs as it is the main topic of the paper”

The title of the review has been changed to “Natural active compounds isolated from essential oils and their antifungal properties”
Part 3 provides an answer concerning compounds of natural origin, the title has been changed to better answer your comment but it seemed adequate to have a more comprehensive aspect before talking about the compounds themselves

There are Tables/Figures lacking in the section related to the mechanism of action

Following your comment, a table 6 has been added to summarize the mechanism of action

“Some tables are not cited in the text”

Indexing was done for tables 1,3,4 and 5 :

Line 38 Table 1

Line  244Table 3

Line 248  Table 4

Line 265 Table 5

The tables provided are quite confusing and should be edited in a different form

The Table 5, which was confusing, has been changed following your advice.

The authors focus too much on French reality, while the paper should give a worldwide broad view on the topic  

Table 1 presents the worldwide epidemiological data of the main invasive fungal infections involved in human pathology

Reviewer 3 Report

Dear,

The work is better compared to the first version, but the 2D structures shown in Table 5 need to be improved for final publication. Another highlight is the writing in item 7 - Example of patents that needs to be improved.

Author Response

Maurine D’Agostino

PhD student at Université de Lorraine

EA SIMPA

54511 Vandoeuvre les Nancy

[email protected]

Nancy, October 10th, 2019

Dear Editor and Reviewers,

Thank you very much for your comments and suggestions. We have been attentive to these comments. Please, find detailed responses to the points you raised. Changes are marked in colours in the text.

Briefly, as suggested by the reviewers, we provided some clarifications for a better understanding of our work.

Your sincerely,

Maurine D’agostino

The work is better compared to the first version,

“but the 2D structures shown in Table 5 need to be improved for final publication.”

The table 5 has been improve according to your advice with formula, 2D and 3D structures from PubChem

“Another highlight is the writing in item 7 - Example of patents that needs to be improved.”

The English of this part has been improved.

Reviewer 4 Report

The authors made the required changes and the manuscript has been significantly improved. However, there are few minor changes:

line 116: Lamiaceae instead of "2.1. Lauriaceae family" table 2: point instead of comma, for example: "Thymol 60,18%" table 2: either Eucalyptol or 1,8-cineole, for consistency table 3: either mg mL−1 or mg/mL, for consistency table 3: the same abbreviation for liter (l or L), for consistency table 4: the latin name of the plants

Author Response

Maurine D’Agostino

PhD student at Université de Lorraine

EA SIMPA

54511 Vandoeuvre les Nancy

[email protected]

Nancy, October 10th, 2019

Dear Editor and Reviewers,

Thank you very much for your comments and suggestions. We have been attentive to these comments. Please, find detailed responses to the points you raised. Changes are marked in colours in the text.

Briefly, as suggested by the reviewers, we provided some clarifications for a better understanding of our work.

Your sincerely,

Maurine D’agostino

The authors made the required changes and the manuscript has been significantly improved. However, there are few minor changes: line 116: Lamiaceae instead of "2.1. Lauriaceae family" table 2: point instead of comma, for example: "Thymol 60,18%" table 2: either Eucalyptol or 1,8-cineole, for consistency table 3: either mg mL−1 or mg/mL, for consistency table 3: the same abbreviation for liter (l or L), for consistency table 4: the latin name of the plants

Thank you for your advices, correction has been made Line 116, table 2, table 3 and table 4.